# Financial and regulatory interventions to reduce unnecessary caesarean sections: An updated scoping review

Rana Islamiah Zahroh[1]*, Alya Hazfiarini[1], Martha Vazquez Corona[1], Thiago Melo Santos[1], Nicole Minckas[1], Newton Opiyo[2], Fahdi Dkhimi[2], Veloshnee Govender[2], Meghan A. Bohren[1], Ana Pilar Betrán[2]

**1** Gender and Women's Health Unit, Nossal Institute for Global Health, Melbourne School of Population and Global Health, The University of Melbourne, Carlton, Victoria, Australia, **2** UNDP/UNFPA/UNICEF/ WHO/World Bank Special Programme of Research, Development and Research Training in Human Reproduction (HRP), Department of Sexual, Reproductive, Maternal, Child, Adolescent Health and Ageing, World Health Organization, 20 Avenue Appia, 1211 Geneva, Switzerland

\* r.zahroh@unimelb.edu.au

## Abstract

Caesarean section (CS) is a life-saving procedure and a critical component of comprehensive obstetric care, yet CS rates are rising globally beyond levels justified by clinical indications. Growing evidence suggests that health system supply-side factors, such as provider payment models that financially reward CS over vaginal birth and the absence or weak enforcement of clinical guidelines, are contributing to this trend. Despite increasing concern, evidence on the implementation and impact of financial and/or regulatory interventions to reduce unnecessary CS remains limited. This scoping review updates and expands a 2020 review by identifying new studies published between 1 January 2019 and 3 September 2024 and synthesising these together with studies included in the earlier review. We searched MEDLINE, EMBASE, CINAHL, Global Index Medicus, and Ebsco MultiDisciplinary Databases, and identified sibling studies to provide additional contextual and implementation details. Across both review periods, we included 46 full-text papers, comprising 31 intervention studies and 15 sibling studies, representing 24 unique interventions. The number of studies has doubled since the 2020 review, with most interventions implemented in high-income countries where baseline CS rates exceeded 20% at the regional or national level. Nearly half of the interventions were financial and complex, integrating multiple context-specific components and primarily targeting hospitals or health workers. Complex regulatory interventions, combining policy mandates with accountability mechanisms, health worker training and guidance, incentives or penalties, women's engagement, and system-level coordination, showed possible benefits for birth outcomes compared with simple financial or regulatory interventions. However, the certainty of evidence was low, and such approaches may be

**Data availability statement:** This study is based on published articles, all of which are either publicly available or accessible through academic institutional credentials. The authors confirm that all other relevant data are included within the main text and/or supplementary files.

**Funding:** This work received funding from the UNDP-UNFPA-UNICEF-WHO-World Bank Special Programme of Research, Development and Research Training in Human Reproduction (HRP), a cosponsored programme executed by the World Health Organization (Award 203444748 to MAB). MAB's time is supported by an Australian National Health and Medical Research Council (NHMRC) Investigator grant (Award 2025634 to MAB). The views expressed in this article are those of the authors and do not necessarily reflect the funders. The funders had no role in study design, data collection and analysis, decision to publish, or preparation of the manuscript.

**Competing interests:** The authors have declared that no competing interests exist.

resource-intensive. Overall, few studies combined financial and regulatory strategies, and maternal and newborn health outcomes were often not assessed, particularly in complex financial interventions. Future research should prioritise context-sensitive, multifaceted interventions and ensure robust evaluation of both service use and health outcomes to avoid unintended harms.

## Introduction

Caesarean section (CS) is a critical life-saving surgical procedure and a key component of comprehensive obstetric care. The World Health Organization (WHO) considers the provision of timely, respectful, and high-quality CS to all women in need as a global priority [1,2]. However, global rates of CS – the proportion of births by CS – are rising beyond levels justified by clinical need [3,4] with increasing use among populations of women who could be considered low-risk for a CS [4,5]. This trend is projected to persist in the coming years [3], raising concerns about the health risks to women and babies, as well as the health system implications, including increased costs and negative impacts on the quality and accessibility of care [6]. Specifically, non-medically indicated CS (referred to as "unnecessary" CS) has been linked to not only adverse maternal outcomes, including severe morbidity, blood transfusion, hysterectomy, and ICU admission, but also neonatal outcomes, such as respiratory complications, NICU admission, and mortality [5,7].

Supply-side health system factors are significant drivers of high CS rates and can often influence women's (through supplier-induced demand) and health workers' (e.g., obstetricians) clinical decisions [8,9,10 11]. For example, payment systems that are output-oriented may incentivise hospitals and health workers to perform more CS, especially where CS is reimbursed at higher rates than vaginal birth [12–14]. The lack of robust audit systems and clinical guidelines on CS, combined with health workers' fears of litigation, may encourage health workers to conduct CS as a defensive practice [13,15]. Furthermore, competing demands, such as practising across multiple hospitals, may influence health workers to favour CS as it is more predictable, and easier to schedule and manage than vaginal birth [12–14].

The influence of the health system and health workers on women's choices and recommendations needs consideration given evidence indicating that most women prefer vaginal births and less than 10% of women actively request CS [10,9]. WHO recommends understanding and addressing these health system factors, especially in low- and middle-income countries (LMICs) where the largest increases in CS rates have been observed [3,16]. This is essential to ensure that CS are performed appropriately, while acknowledging and addressing the power imbalances and hierarchical relationships that exist between health facilities, health workers, and women. [17].

Several reviews have explored health system interventions aimed at reducing unnecessary CS. Chapman et al (2019) [18] and Chen et al (2018) [19] evaluated interventions targeting health workers and health organisations, such as audit and feedback, second opinion for CS decision, and peer review; both found

inconsistent effects in reducing CS rates. Boatin et al (2018) [20] assessed audit and feedback interventions based on the Robson classification, reporting limited evidence of effectiveness. Torloni et al (2023) [21] assessed interventions to increase or reintroduce assisted vaginal birth (forceps or vacuum), highlighting feasibility challenges and the need for skill retention and training. Only two systematic reviews, by Yu et al (2019) [22]and Opiyo et al (2020) [8], evaluated both financial and regulatory interventions published up to 2018 and 2019, respectively; both found mixed effects and generally low-quality evidence [8,22]. More rigorous studies and innovative methods for assessing the impact of financial and regulatory interventions were deemed necessary on CS rates, safety, and cost-effectiveness [8,22]. In the five years since these previous reviews were published, new evidence may now be available to address these research gaps.

We aimed to update Opiyo et al (2020) [8] scoping review on financial and/or regulatory interventions implemented globally to reduce unnecessary CS, by synthesising new evidence published since the 2020 publication. The specific objectives were to: (i) describe the characteristics of identified financial and/or regulatory interventions intended to reduce unnecessary CS, and (ii) explore the effectiveness, safety, and related outcomes of the identified financial and/or regulatory interventions.

## Materials and methods

This scoping review adhered to the Preferred Reporting Items for Systematic Reviews and Meta-analysis extension for scoping reviews (PRISMA-Scr; S1 Appendix) [23]. The protocol is registered with the Open Science Framework (https://osf.io/7btgp).

### Criteria for considering studies for this review

**Type of studies.** All randomised trials, non-randomised trials, before-after studies, interrupted time-series studies, cohort studies, large-scale intervention case studies, implementation studies, pre-post intervention studies, cross-sectional, and mixed-methods studies were eligible for inclusion. We included studies published in any language. We excluded non-primary papers (e.g., case reports, letters, editorials, commentaries, reviews, study protocols, posters, and conference abstracts).

**Types of participants.** We included studies involving groups of low-risk pregnant women, health workers, and health facilities providing maternity care. Low-risk pregnant women were defined as those receiving antenatal and intrapartum care in health facilities, with term, singleton, and cephalic pregnancies, with or without a previous CS, corresponding to Robson groups 1–4 [24]. However, we also included any studies involving women in Group 5 (those with a previous CS), as together Groups 1–5 represent the majority of women giving birth and are the main contributors to the rising CS rates globally [4]. If the Robson classification was unclear or not explicitly defined, studies were included if the population was likely to pertain to low-risk women. We excluded studies that involved exclusively women with specific medical conditions (e.g., diabetes, obesity, or HIV). Health workers included nurses, midwives, obstetricians, and doctors involved in antenatal or intrapartum care. We included studies conducted in health facilities with childbirth care (e.g., hospitals, birthing centres).

In this scoping review, we use the term "women" to refer to individuals with reproductive capacity for pregnancy and childbirth. We acknowledge that not all people who give birth identify as women and recognise the importance of inclusive language. However, we have chosen to retain the term "women" in this context to highlight the gendered systemic discrimination that women face in maternity care. Gendered expectations, medicalisation, and structural inequities disproportionately affect women's experiences and access to respectful, evidence-based care. While this does not diminish the experiences of gender-diverse pregnant people, the decision to use "women" reflects the historical and ongoing challenges in maternal health policy, practice, and research.

**Settings.** We included studies conducted in any country, at any health system level (e.g., national, regional, district) and any type of health facilities (e.g., hospitals, birthing centres, public, private).

**Type of interventions.** We included studies reporting any financial and/or regulatory interventions, including complex interventions that were either multi-component (comprising multiple interrelated elements within a single intervention) or multi-faceted (targeting different levels such as women, health workers, health facilities, or health systems). Interventions were eligible if they included financial and/or regulatory measures and were aimed at reducing unnecessary CS. We excluded studies aiming to increase CS use because interventions intended to reduce unnecessary procedures reflect growing concern over the global oversupply of CS, associated health risks, and the strain on health systems [6,17].

We defined financial and regulatory interventions according to the WHO health system building blocks (S2 Appendix). Financial interventions refer to the implementation of a strategy or reform in healthcare financing, including those designed to influence service delivery and reduce unnecessary CS. This includes changes in the way health providers are selected and contracted, how and how much they are paid for these services, and what reporting is required in exchange for such payments. Regulatory interventions refer to the implementation of policies, regulations, or legislation, including initiatives to strengthen the role of patients or communities in health governance, designed to reduce the unnecessary use of CS.

**Type of outcome measures.** All outcomes related to the effectiveness (i.e., reduction of CS, repeat CS, elective) or safety (i.e., maternal and neonatal health outcomes) of the intervention reported in the included studies were extracted, mapped, and reported.

**Search methods for identification of studies.** We searched MEDLINE, EMBASE, CINAHL, Global Index Medicus, Ebsco MultiDisciplinary Databases to identify eligible studies (S3 Appendix). We used the search strategy from Opiyo et al (2020) review [8] for each database, and searched for studies published between 1 January 2019–3 September 2024. However, to get more comprehensive picture on the state of financial and regulatory interventions globally, our analysis also included studies from Opiyo et al (2020) review. Search terms included terms related to CS and relevant financial and regulatory interventions. As we aimed to include complex interventions involving financing and/or regulatory measures, we re-screened all full-text papers excluded by Opiyo et al (2020) [8] to identify any studies that might meet our broader inclusion criteria. The WHO International Clinical Trials Registry and ClinicalTrials.gov were also searched for any ongoing studies.

We searched for sibling studies of the included interventions for additional information on the characteristics of the interventions or additional outcomes that may not have been presented in the papers retrieved by our searches. Sibling studies were defined as publications linked to the same intervention as the main study but reported separately. These included, for example, formative research, process evaluations, costing analyses, and cost-effectiveness studies. To locate sibling studies, we reviewed reference lists of all included intervention studies, and conducted a forward reference search using the "Cited by" function in PubMed and Google Scholar. One review author screened potential sibling studies, which a second review author double-checked. Disagreements were discussed and adjudicated by the third review author if needed.

**Selection of studies.** The titles and abstracts retrieved from various databases were compiled into a single reference database using EndNote (www.endnote.com), where duplicates were removed. Following deduplication, all references were imported into Covidence (www.covidence.org). We duplicated the Covidence project from the Opiyo et al (2020) [8] review, to not re-screen studies already included in their review. Two independent review authors assessed study eligibility based on the title and abstract against predefined inclusion and exclusion criteria. Studies meeting the inclusion criteria underwent full-text retrieval for further assessment. Studies that did not meet the inclusion criteria were excluded. In cases of disagreement during any stage of screening and assessment, consensus was sought through discussion among the research team.

**Language translation.** During full-text screening, we found two studies in Portuguese, two studies in Persian, and one study in Mandarin, which were translated to English using an open-source software, Google Translate, to verify the eligibility. If eligible for inclusion, the full text and the extraction were double reviewed by a native speaker.

**Data extraction.** We developed a specific form for data extraction for this review, including information on study objectives, methodology, design, settings, participant characteristics, and description of intervention (See S4 Appendix for data extraction details). The data extraction was pilot-tested by three reviewers on three studies to ensure accuracy and consistency, and reviewed by all authors to provide feedback before final use. One author conducted the primary data extraction, which was then independently cross-checked by another author. Any discrepancies were discussed and resolved through consensus with the wider author team. This process helped to ensure that the data extraction was both sound and appropriate for the review objectives.

**Assessing the quality of included studies.** The Cochrane Risk of Bias In Non-Randomized Studies of Interventions (ROBINS-I) tool was used to appraise the quality of the included studies. Domains assessed for risk of bias included confounding, participant selection, intervention classification, adherence to intended interventions, missing data, outcome measurement, and selective reporting.

After the risk of bias assessment, two reviewers evaluated the confidence or certainty of the body of evidence for each outcome (referred to as "quality of evidence" or "confidence in the estimate") using the Grading of Recommendations Assessment, Development and Evaluation (GRADE) framework. According to this method, the certainty of evidence for each outcome is graded as "High", "Moderate", "Low", or "Very low" based on predefined criteria. Any disagreements between the two reviewers were resolved through discussion with a third reviewer.

GRADE assessments are not standard practice in scoping reviews; however, consistent with methodological guidance that scoping reviews may include optional critical appraisal where it adds value [25]. We applied GRADE in a descriptive and interpretive manner. The GRADE assessments were used only to contextualise and inform interpretation of the findings, not to exclude studies or formally rank the evidence. The results of the risk of bias and certainty assessments are presented in S5 Appendix.

**Data analysis and synthesis.** To enable comparison across studies, we categorised all extracted data according to intervention settings using the World Bank income level [26], WHO region classification, and health system level (national, regional, and facility). We further classified intervention characteristics using WHO Health System Building Blocks [27], the target of the intervention (women, health workers, health organisations), and the type of intervention (financial and/or regulatory).

We also categorised interventions as either simple or complex based on adapted criteria from two WHO frameworks: 1) the WHO strategic health purchasing framework to classify health financing interventions [28], and 2) the WHO frameworks on health system governance for universal health coverage and primary healthcare operational to classify regulatory interventions [29,30]. These frameworks define different types of elements in financial and regulatory interventions, as outlined in Table 1. The framework in Table 1 was used as a guide for classifying interventions into simple or complex, recognising that a single intervention can include multiple elements (e.g., benefit design and payment methods).

Interventions were classified as complex if they implemented more than one type of element within a single intervention. For example, shifting from fee-for-service to a global budget was a "simple financing intervention" as it involved only a payment method change element per Table 1. In contrast, an intervention that combined laws to enforce clinical guidelines, hospital review boards, and mandatory reporting was a "complex regulatory intervention", as it involved several elements of regulatory interventions per Table 1, (e.g., formulating policy and strategic plans, improving quality care, and ensuring accountability).

We also categorised each reported outcome based on the classifications used in another review [31] to determine the level of impact of each intervention based on effect estimates and certainty of evidence. Outcomes were categorised as "possible benefit", "possible no difference of effect", or "possible harm" based on the reported effect estimates and

confidence intervals. For example, a reduction in CS rates or improved maternal satisfaction was considered a "possible benefit", whereas no significant change in maternal mortality was classified as "possible no difference of effect". Interventions associated with increased complications, such as higher rates of postpartum haemorrhage, were classified as "possible harm". Table 2 summarises the outcomes, effect estimates, and certainty of evidence corresponding to each category. The type of outcomes and their impact classification for each study can be seen in S6 Appendix.

**Table 1. Types of elements in financial and regulatory interventions.**

| Types of elements in financial interventions |
|---|

**Benefit design**
Any strategies that modify the conditions of access attached to publicly funded healthcare benefits. For example, any changes in copayment or care pathway (e.g., Enforcing referral).

**Provider selection processes**
Any strategies that focus on selecting the type of providers that are eligible to provide the service. For example, any form of provider selection process (limiting access to public providers, or to a narrower set of providers).

**Contracting modalities including reporting obligations**
Any strategies that focus on the relationship between payers (e.g., government or insurance bodies) and providers. Examples include mechanisms such as audits, performance targets, and reporting requirements, aimed at enhancing accountability and service quality.

**Payment methods**
Any strategies that implement the way (how much, how, and when) providers are compensated, aimed at influencing their behaviour and the efficiency of service delivery. Payment methods (e.g., capitation, fee-for-service, case-based payment) are designed to balance cost control.

| Types of elements in regulatory interventions |
|---|

**Formulating policy and strategic plans**
Any strategies defining governance structures for priority-setting, policy, or legislative mandate, to ensure evidence-based decision-making. Examples include setting national health strategies, policies, guidelines, and protocol implementation.

**Generating intelligence**
Any strategies designed to produce information for decision-making to improve clinical practice, such as review mechanisms or other activities, as far as the endpoint is to produce information to improve care.

**Putting in place levers or tools for implementing policy**
Any strategies or activities to address policy barriers and bottlenecks, such as the implementation of incentives or penalties, recognition programs or financial rewards/penalties based on governance targets.

**Ensuring accountability**
Governance structures or strategies to improve accountability by establishing reporting obligations, setting clear roles and responsibilities, monitoring and evaluation. Example includes the establishment of review boards, agencies or coalitions, regular audit and feedback, peer performance scorecards, publicly accessible CS rates dashboards to track progress, information systems, or other activities to improve accountability.

**Improvement of the quality of care**
Any strategies for implementing a service or program to improve quality of care that is safe, effective, people-centred, timely, efficient, equitable, and integrated. Which may target any of the following:
• *Health system environment:* capacity building for health workers, such as training or mentoring, or improvement of physical infrastructure, such as enhancing spaces to conduct vaginal births
• *Reducing harm:* implementation of new care models, or activities that uphold foundational principle of causing non avoidable harm to people
• *Improvement in clinical care:* implementation of hospital-based clinical pathways, protocols, context-appropriate standards, checklists, standardised clinical forms and clinical decision support tools
• *Patient, family and community engagement and empowerment:* involvement of relevant stakeholders, including women and their families

After the categorisations were completed, the data were systematically summarised in a descriptive format, qualitatively assessing intervention characteristics and reported outcomes. Findings were synthesised to identify common themes, patterns, and variations across studies. When the same intervention was reported in multiple journal articles or reports, these were merged into a single study, with the unique study being the primary unit of review rather than each report. Furthermore, if multiple studies assessed the same intervention, but reported the components differently, we reported all the components together to get a comprehensive view of the intervention.

## Result

### Overview of included studies

We identified 4,272 titles and abstracts from database search updates and trial registry searches. After eligibility screening, we included a total of 46 full texts from 31 intervention and 15 sibling studies, reporting on 24 unique interventions (Fig 1). No ongoing clinical trials were found.

All studies aimed to evaluate the impact of policies on CS rate reduction (n=31/31, 100%). Forty-five per cent commenced between 2011 and 2021. The most common study designs were retrospective before-and-after studies (n=14/31; 45.2%) and interrupted time series (n=9/31; 29.0%). The included studies were mostly conducted in high-income countries (n=20/31; 64.5%), with the majority of studies conducted in the United States of America (USA) (n=8/31; 25.8%), Iran (n=7/31; 22.6%), and Taiwan (n=5/31; 16.1%). More than half of the studies analysed >100,000 births (n=18/31;58.1%). Almost half of the studies analysed data in less than 5 (n=6/31;19.4%) or between 10–50 (n=4/31;12.9%) health facilities. The most common type of health facility was either a mix of private and public (n=14/31;45.2%) or public (n=7/31;22.6%). An overview of the included studies is shown in Fig 2, and detailed characteristics can be found in S7 Appendix.

Fig 3 shows an overview of the 24 included interventions. The types of interventions implemented were either complex regulatory (n=6/24;25.0%), complex financing (n=6/24;25.0%), or simple financing interventions (20.8%;5/24). Most interventions targeted the health financing building blocks (n=11/24;45.8%). Almost 80% (n=19/24) of the interventions were implemented in settings where baseline CS rates were between 20% and 78%. Most interventions were implemented at either the regional (n=12/24;50.0%) or national levels (n=10/24;41.7%), and more than half targeted either health organisations (n=7/24;29.2%) or health workers (n=6/24;25.0%). Only two interventions adopted theoretical or behaviour change frameworks during intervention design (Institute for Healthcare Improvement Scale-Up Framework and

**Table 2. Outcome classifications adapted by [31].**

| Category | Effect estimate (RR/OR/WMD)* | Certainty of evidence** | Terminology in text |
|---|---|---|---|
| Possible benefit | Evidence or effect of benefit with and without confidence interval (if with, the 95% confidence interval should not cross the line of no effect) | Low | "May reduce/increase… " |
| Possible no difference of effect | Evidence or effect near the line of no effect with and without confidence interval (if with, a narrow 95% confidence interval crossing the line of no effect between 1) | Low | "May have no effect …" |
| Possible Harm | Evidence or effect of harm with and without confidence interval (if with, the 95% confidence interval not crossing the line of no effect) | Low | "May reduce/increase …" |
| No conclusion possible | Any evidence or effect estimates with a wide 95% confidence interval crossing the line of no effect substantially | Low | "There is insufficient evidence …" |
| | Any evidence or effect estimates | Very low | "It is uncertain whether …" |
| No data | No studies or no data | Not applicable | "No studies …" |

*RR: risk ratio, OR: odds ratio, WMD: weighted mean difference.

**All included studies were observational or non-randomised. Under the GRADE framework, observational studies begin as low-certainty evidence and can only be upgraded when findings are consistent across multiple studies or demonstrate very large effects [32,33]. As each intervention was reported in a single study, certainty could not be upgraded; therefore, all ratings presented here reflect low or very low certainty, consistent with GRADE guidance.

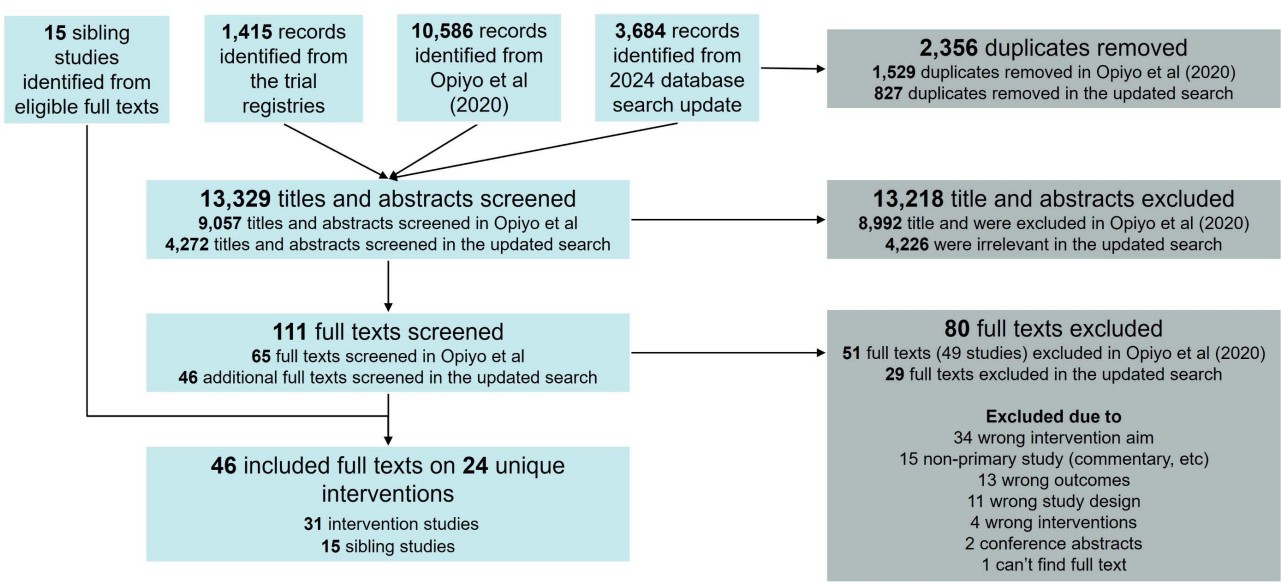

**Fig 1. PRISMA flowchart.**

the Consolidated Framework for Implementation Research). Prior to applying the risk of bias and certainty of evidence criteria (Table 2), 54% (n = 13/24) of interventions reported a CS reduction, while the remaining showed either an increase (n = 7/24;29.2%), no change (n = 2/24;8.3%), or mixed effects (n = 2/24;8.3%). Most studies had a "not serious" risk of bias (n = 20/31;62.5%), while 28.1% were rated as "serious" (n = 8/31) and 9.4% as "very serious" (n = 3/24). Due to the heterogeneity across studies and the inability to pool the data, the certainty of evidence was either "low" (n = 22/31;71.9%) or "very low" (n = 9/31;28.1%) for most studies. See S5 Appendix for more details.

### Financial interventions

**Simple financial interventions.** Simple financial interventions primarily modified provider reimbursement structures to reduce financial incentives favouring CS over vaginal birth, without introducing additional regulatory or quality oversight components. In practice, these interventions involved either replacing fee-for-service payments with blended or bundled payments, or equalising reimbursement levels for CS and vaginal birth by adjusting one or both payment rates.

Across five studies, findings were mixed and of low certainty. Two studies from the USA and China evaluating shifts from fee-for-service to blended or episode-bundle payments reported possible benefits in reducing CS rates [34,35]. In contrast, three studies focusing solely on fee equalisation reported inconsistent findings [36–38]. One-tailed equalisation (raising vaginal birth fees only to match CS) produced mixed results in Italy and Taiwan [36,38], with only the Italian study demonstrating possible benefit in CS reduction within a pre-existing Diagnosis-Related Group (DRG) system [36]. By comparison, a two-tailed approach (raising vaginal birth fees while lowering CS payments) in the US showed possible benefits in reducing CS rates [37]. Overall, reported reductions in CS rates were observed primarily in studies where financial changes were implemented within structured payment systems, rather than through isolated fee adjustments.

**Complex financial interventions.** Complex financial interventions combined structural changes to provider payment systems with additional administrative or accountability components, aiming to influence health workers' clinical decision-making

PLOS Global Public Health

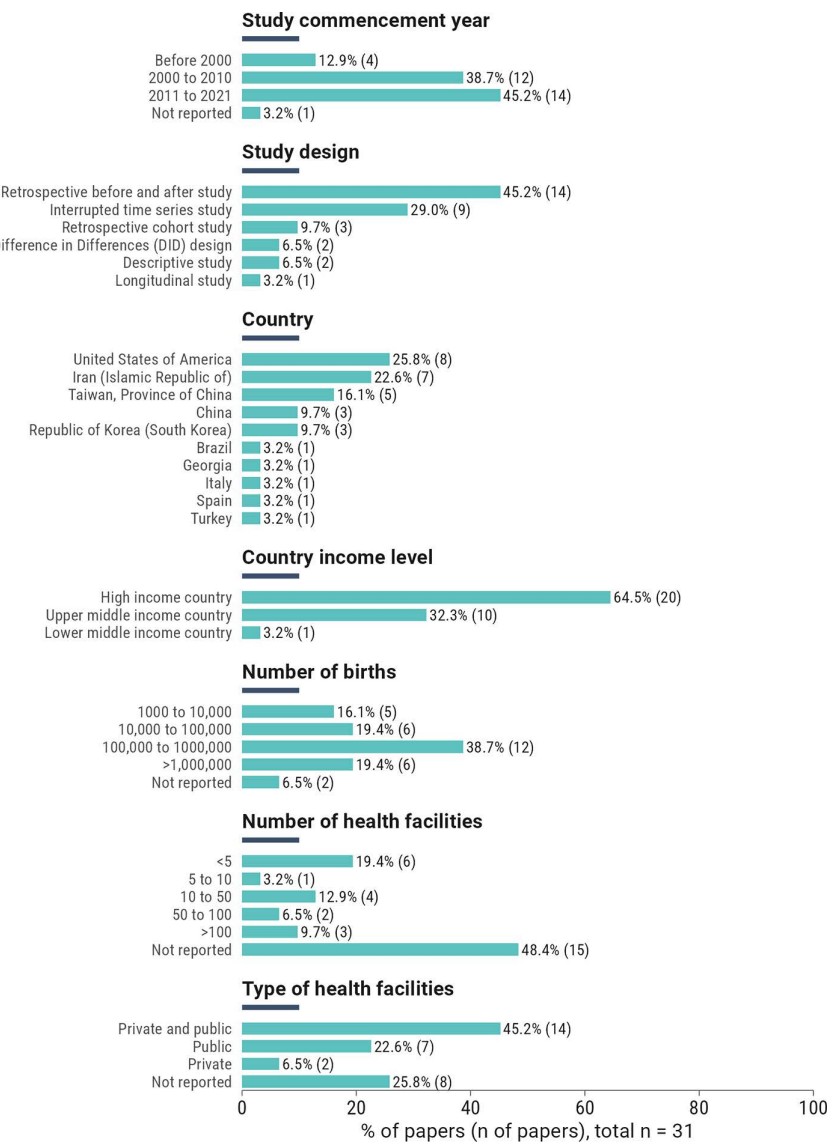

**Fig 2. Overview of the 31 included intervention studies.**

through financial and organisational mechanisms. In practice, these interventions replaced fee-for-service payment with global budgets, DRG-based prospective payments, capitation, or case-based reimbursement, accompanied by audit, peer review, or performance-linked incentives.

Across seven studies, reported effects were mixed and generally low-certainty evidence. In Taiwan, two studies evaluating national and facility-level shifts from fee-for-service to a global budget payment system [39,40], incorporating one-tailed fee equalisation [39], co-payments by women for non-medically indicated CS [39], post-CS peer reviews [40], and audit and feedback [40], reported possible no difference in CS rates.

DRG-based payment reforms in Korea yielded inconsistent findings: one study reported a possible benefit in CS reduction [41], while another found no change [42]. Introducing a capitation-based system alongside the removal of

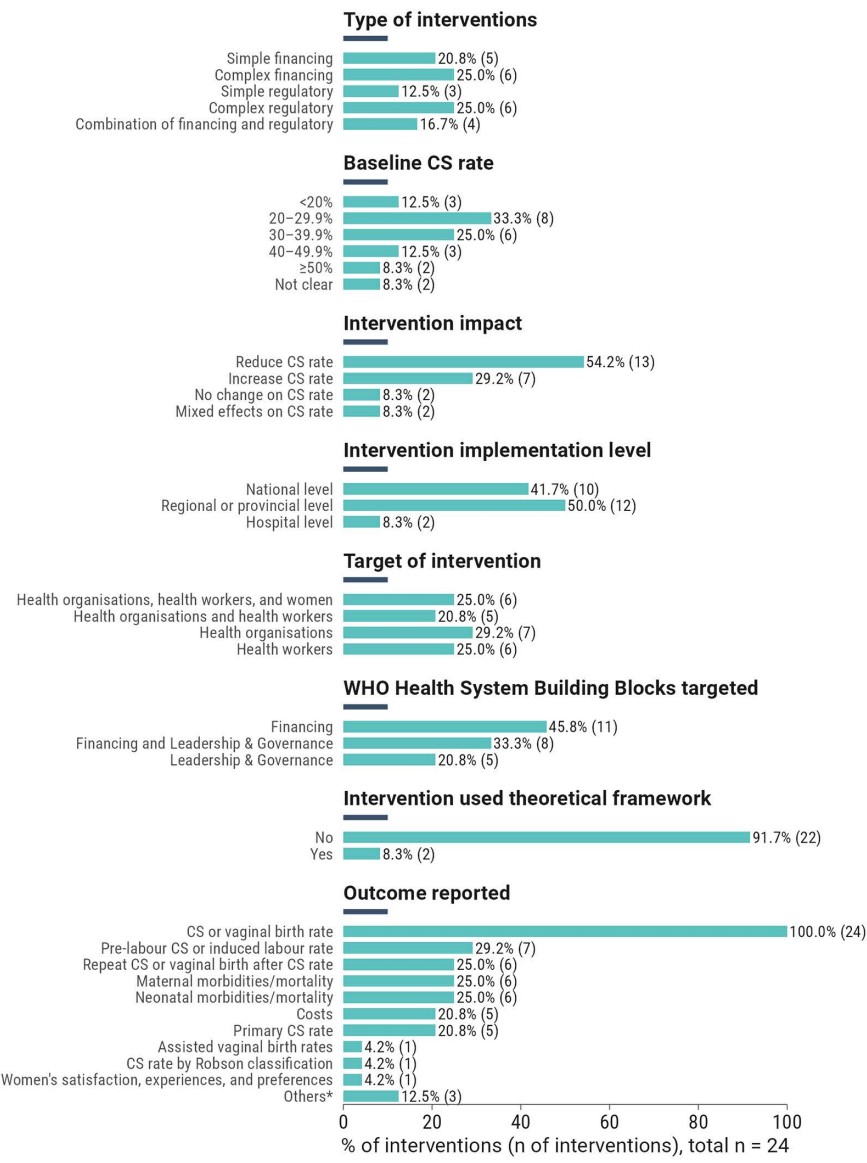

**Fig 3. Overview of 24 unique interventions in included studies.**

co-payments by women in China [43] and limiting provider selection under managed care in the USA [44] were both associated with possible increases in CS rates. In Taiwan, a case-based payment model with performance incentives reported a reduction in CS rates, though the evidence was very low certainty, and the effect remains uncertain [45].

Overall, complex financial reforms showed mixed and uncertain effects across payment models: DRG-based systems demonstrated possible but inconsistent reductions in CS; case-based payments appeared potentially promising but with very low certainty.

### Regulatory interventions

**Simple regulatory interventions.** Simple regulatory interventions involved policy changes that directly constrained clinical decision-making or modified the legal environment in which obstetric care was provided, without additional

enforcement or quality improvement components. Three studies from the USA evaluated such interventions. Two studies examined labour induction policies: one mandating universal offering of elective induction at 39 weeks [46] and another introducing a state-wide "Hard Stop" to restrict elective inductions and CS before 39 weeks [47]. Both reported possible reductions in CS rates, although evidence certainty was low.

A third study assessed state-level damage cap policies to limit health workers' liability in malpractice litigation [48], but findings were inconclusive due to very low certainty. Overall, regulatory policies that directly restricted the timing or permissibility of elective procedures were more commonly associated with possible reductions in CS rates than legal liability reforms, although the small number of studies and low-certainty evidence limit firm conclusions.

**Complex regulatory interventions.** Complex regulatory interventions combined formal policy mandates with enforcement and multi-component quality improvement strategies, aiming to influence clinical practice through accountability, support, and system-level coordination. Six studies evaluated such interventions. Possible benefits in reducing CS rates were reported in studies from China [49], Georgia [50], USA [51,52], and Brazil [53], though the certainty of evidence was low. These successful studies, implemented policy mandates (CS rate thresholds, signed informed consent requirements for maternal request by CS), accountability mechanisms (audit and feedback, public reporting), health worker support (training, guidelines), incentives or penalties (financial or recognition-based), women's engagement (birth plans, decision-making, prenatal education, birth companions), and system-level coordination (multi-stakeholder collaboration, integrated quality improvement).

In contrast, one study in Spain [54], evaluated a policy promoting vaginal birth that focused on updated protocols, training, infrastructure support, and women's involvement but lacked explicit accountability mechanisms, reported a possible increase in CS rates; findings were inconclusive due to very low certainty of evidence.

Overall, complex regulatory interventions that combined policy mandates with accountability mechanisms, health workers' training and guidance, incentives or penalties, women's engagement, and system-level coordination were more commonly associated with possible reductions in CS rates, whereas interventions focusing solely on service improvements without accountability mechanisms showed no consistent benefit.

## Combination of financial and regulatory interventions

We identified only two types of combined interventions: (1) simple regulatory and simple financing interventions, and (2) complex regulatory and simple financial interventions. We did not find any interventions that combine complex financing with simple regulatory interventions.

**Simple regulatory and simple financing interventions.** Simple combinations of regulatory and financial interventions linked basic policy targets with direct financial consequences for health workers or hospitals, without additional enforcement or quality improvement components. Three studies evaluated such interventions.

In Turkey, a Ministry of Health-issued policy linked health workers' compensation to CS rate, providing full payment when health workers' CS rates ≤15% and reduced payment above this threshold [55]; reporting a reduced CS rate, but could be inconclusive due to very low certainty. In contrast, in Korea, the Value Incentive Program, which provided 1–5% fee increase for high-performing hospitals and a 1–5% fee reduction for underperforming hospitals, was associated with increased CS rates (low certainty) [56]. Similarly, in Taiwan, a policy combining cap of 30% CS rates, hospital accreditation requirements, and near zero-cost childbirth option for women was also associated with a possible increase in CS rates (low certainty) [57]. Overall, simple combinations of regulatory and financial measures showed mixed effects, with only one study suggesting a possible reduction in CS rates, yet with very low certainty, and others showing increases.

**Complex regulatory and simple financial interventions.** Complex regulatory and simple financial interventions combined multi-component regulatory reform with limited financial measures, aiming to influence CS rates through policy mandates alongside direct financial incentives or cost removal. Seven studies evaluated the Iranian Health Transformation Plan, which set the CS rate thresholds, provided bonus payments for health workers, removed co-payments for women, and improved maternity infrastructure and services.

Across more than 100 health facilities, outcomes were mixed: two studies reported possible benefits in reducing CS rates [58,59], one reported possible harm in increasing CS rates [60], one found no change [61], and three were inconclusive [62–64], all with low or very low certainty. Overall, the Iranian Health Transformation Plan demonstrated inconsistent and context-dependent effects on CS rates, with substantial variation across facilities and no consistent pattern of reduction observed.

## Intervention outcomes

Outcomes reported included overall CS rates (n = 24/24;100%), rates of different types of CS or vaginal birth (pre-labour CS, induced labour, vaginal birth after CS, or repeat CS; n = 13/24, 54.2%), and maternal or neonatal morbidities and mortality (n = 12/24;50%). Only five studies reported cost-related outcomes (n = 5/24;20.8%), one study (n = 1/24;4.2%) reported assisted vaginal birth outcomes, one study (n = 1/24;4.2%) reported CS rates by Robson classification, and one study (n = 1/24;4.2%) reported women's experiences of and perceptions about CS (Fig 4).

Fig 4 summarises intervention types alongside the direction of effect on CS rates and related outcomes. Overall, studies evaluating complex regulatory interventions assessed a wider range of outcomes and tended to report more possible benefits related to mode of birth and maternal and neonatal health, compared with other intervention types. However, given the overall low to very low certainty of the evidence, these findings should be interpreted with caution. Among the six studies of complex regulatory interventions, several reported possible reductions in overall CS rate (5/6 studies; [49–53]), repeat CS (1/6 study; [52]), and elective CS (1/6 study; [49]), as well as possible increases in induced labour (2/6 studies; [50,53]), vaginal birth (1/6 study; [53]), and assisted vaginal birth rates (1/6 study; [50]). Some studies also suggested possible benefits for maternal and neonatal outcomes, including reduced maternal mortality (1/6 study; [53]), higher maternal satisfaction (1/6 study; [53]), lower infection rates (1/6 study; [53]), reduced neonatal mortality (1/6 study; [53]) and more babies born at term and with higher birth weights (1/6 study; [53]). Possible harms from increased neonatal respiratory complications were reported in one study [53]. Only one study examined cost outcomes, finding that the intervention was expensive and not cost-effective [53].

Simple regulatory interventions also showed possible benefits in reducing overall CS (2/3 studies; [46,65]) and elective CS (1/3 study; [47]). However, these interventions were also associated with possible harms to maternal and neonatal outcomes (Fig 4), such as postpartum haemorrhage (2/3 studies; [46,47,65]), peripartum infection (2/3 studies; [46,65]), and neonatal respiratory complications (1/3 studies; [65]). Two policies for induction of labour appeared to reduce CS rates but also showed potential harm: offering universal elective induction at 39 weeks gestation [46], and restricting elective induction before 39 weeks gestation [47].

Simple financing interventions also showed possible benefits in reducing overall CS rates (3/5 studies; [34–36]), with one study revealing possible benefits in reducing hospitalisations. However, one study on blended payment found possible harm in the form of increased risk of postpartum haemorrhage [34]. Evidence on cost-related outcomes was mixed, with two studies suggesting possible benefits [34,35] and another indicating possible harms [36]. Overall, there are limited studies that combine financial and regulatory interventions aimed at reducing unnecessary CS, and safety outcomes were often not evaluated in complex financial interventions.

## Discussion

The number of studies evaluating financial and/or regulatory interventions has doubled since the publication of the Opiyo et al (2020) [8] review, indicating growing interest in innovation to address unnecessary CS. Most studies used retrospective before-and-after designs, focused on financial reform policies, and were often complex interventions implemented in high-income countries where baseline CS rates exceeded 20%. These interventions typically targeted either health organisations or health workers. Complex regulatory interventions combining policy mandates with accountability mechanisms,

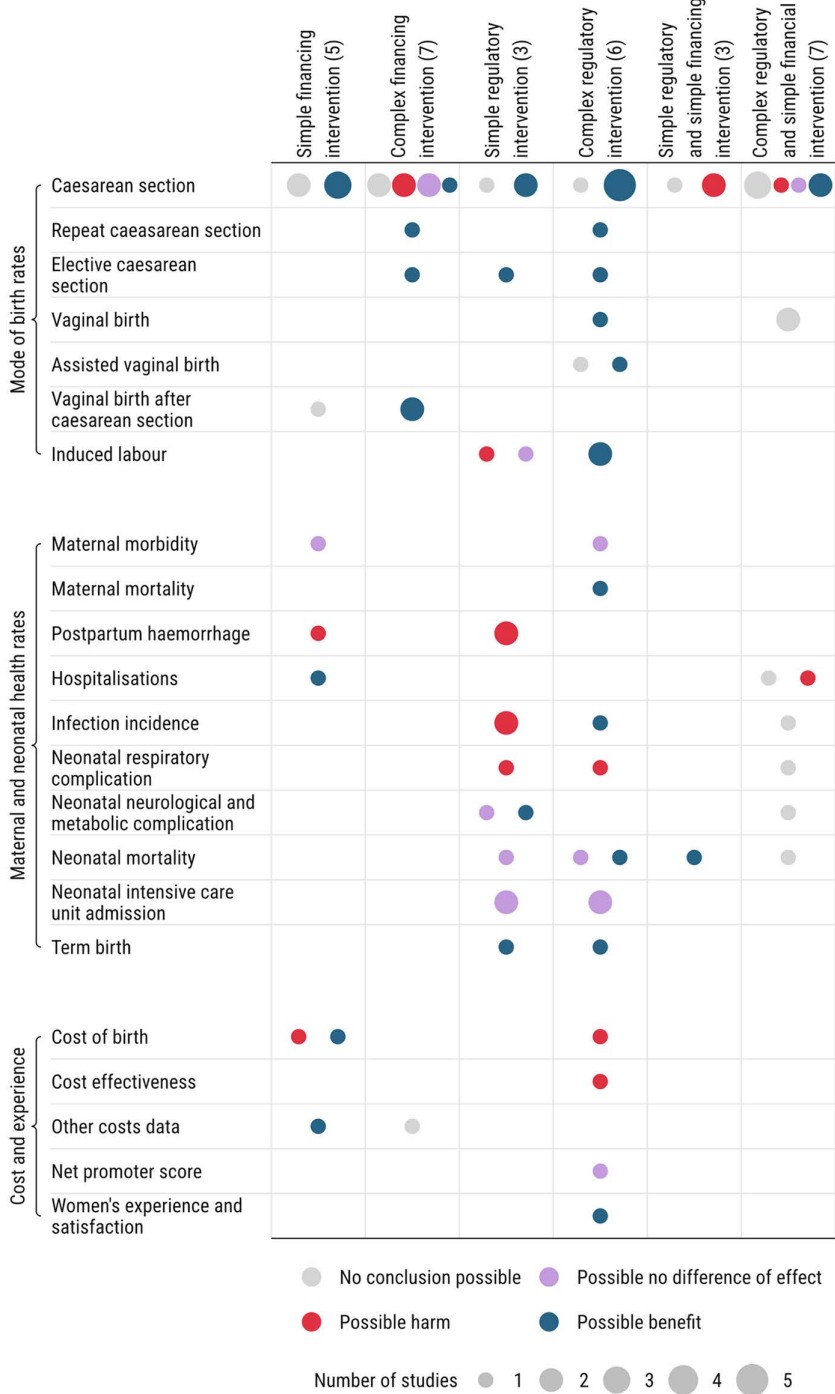

**Fig 4. Overview of intervention types mapped against direction of effect on CS rates and related outcomes.**

health workers' training and guidance, incentives or penalties, women's engagement, and system-level coordination tended to report more possible benefits to reduce CS rates and ensure safety. However, these findings were based on low-certainty evidence, and such interventions were often resource-intensive. Simple and financial interventions that have

structured-payment reform at the system level, like DRG, also suggested possible reductions in CS rates, but evidence is varied across contexts and generally lacks data on safety. Overall, few studies have assessed combined financial and regulatory approaches, and evidence remains limited on their cost-effectiveness and safety impacts. This review advances current knowledge by updating and expanding the evidence base since Opiyo et al (2020) [8] review, providing a more detailed typology of financial and regulatory interventions, highlighting emerging patterns of effectiveness and safety, and identifying persistent gaps in cost-effectiveness, implementation, and evidence from LMICs.

Reducing unnecessary CS requires multi-component and multi-faceted interventions, as the factors driving high CS rates are complex and interconnected across individual, facility, and health system levels [17]. Our findings align with the Lancet CS Series (2018) [17] and the Cochrane review of non-clinical interventions [19], both of which emphasise that the determinants of CS use and the effectiveness of interventions are shaped by multi-level influences spanning women's preferences, health worker practices, institutional culture, and wider health system structures. Consistent with these studies, our review found that complex regulatory interventions, those combining multiple coordinated strategies across policy, facility, and community levels, were the ones that showed possible benefits in reducing CS rates. Single-component interventions may fail to achieve or sustain reductions because they often overlook the broader organisational, cultural, and contextual factors shaping clinical decision-making. Evidence from prior studies also supports the need for integrated approaches, combining several coordinated strategies within one intervention, to promote and sustain change [66,67]. This is aligned with the WHO statement that "multi-faceted (rather than single-component) interventions tailored to local determinants (barriers and facilitators) of CS practices are recommended" to reduce unnecessary CS [68].

Key complex regulatory interventions to reduce unnecessary CS identified in this review include the implementation of **policy mandates** (CS rate thresholds, signed informed consent requirements for maternal request by CS), **accountability mechanisms** (audit and feedback, public reporting), **health worker support** (training, guidelines), **incentives or penalties** (financial or recognition-based), **women's engagement** (birth plans, decision-making, prenatal education, birth companions), and **system-level coordination** (multi-stakeholder collaboration, integrated quality improvement). This finding is consistent with recent evidence [66] that identified training, clinical protocol, actionable audit and feedback, multidisciplinary collaboration, and providers' willingness to change are key components of successful interventions targeting health workers to optimise CS. Our review expands this evidence by showing that women's active engagement, through involvement in birth planning, informed decision-making, companionship or respectful maternity care, may further enhance the effectiveness of system-wide efforts to reduce unnecessary CS. The practice of establishing CS rate thresholds and imposing penalties or incentives for health workers who are over or under the threshold may create strong accountability mechanisms that encourage adherence to guidelines and motivate health workers to make evidence-based decisions [29,30]. However, these practices could also result in unintended consequences, such as health workers opting for vaginal births when a CS is clinically indicated, compromising the quality of care and safety of women and babies [50].

This review highlights the importance of anticipating and monitoring potential unintended effects on women's and babies' safety when implementing interventions to reduce CS rates. Several studies evaluating simple regulatory and simple financial interventions reported increases in adverse maternal outcomes (postpartum haemorrhage, maternal blood transfusion, chorioamnionitis) when CS was reduced [34,46,65]. These adverse outcomes may be attributable to longer or more complex labours as care shifts away from CS, particularly in cases of labour induction or attempts for vaginal birth in borderline or high-risk scenarios [34,46,65]. They may also arise from improved detection or reporting following changes in clinical protocols [34,46,47,65]. These findings highlight that a blunt reduction in overall CS rates, while appealing as a policy target, can be unsafe if it does not distinguish between medically necessary and unnecessary CS. Without adequate safeguards, such interventions risk inadvertently restricting access for women or babies with medically necessary CS, rather than reducing unnecessary CS. A safe and equitable CS reduction requires improved management of labour, audit and feedback mechanisms, and continuous monitoring to ensure that quality and timely access to medically indicated CS are not compromised [2].

Our review shows that system-level financial reforms from fee-for-service to certain payment systems, such as DRG, tend to show greater possible benefits in reducing CS rates than simply equalising vaginal birth and CS fees. This finding aligns with a related review [22], which reported that simple economic incentives alone were often insufficient to influence health workers behaviour. Together, these findings suggest that financial interventions must be substantial, system-oriented, and adaptive to drive meaningful change. It is unlikely that a single payment mechanism alone can achieve sustained reductions in unnecessary CS. Effective purchasing arrangements require on-going adjustment of payment methods, rates, and contracting modalities, to reflective evolving system goals. The experience from Taiwan offers a powerful example of this. Initially, Taiwan equalised payment rates by aligning the fees for vaginal births with those for CS [38,45,57], which was effective in reducing CS rates. However, growing concerns over higher health expenditure later prompted a shift toward cost-containment measures [39,40]. This sequence demonstrates how policies achieving one objective can inadvertently create new challenges, underscoring the need for continuous monitoring and course correction. At the same time, the scale and timing of reforms matter. While policies must remain responsive to emerging issues, frequent or poorly timed adjustments can disrupt implementation and reduce effectiveness. Policymakers should therefore seek a balance between adaptability and stability when designing and refining reform efforts.

### Implications for practice and research

To strengthen the evidence for reducing unnecessary CS, future research should prioritise high-quality studies evaluating the effectiveness of financial and regulatory interventions. More rigorous designs are needed to establish causality, assess long-term outcomes, and capture contextual factors such as overall CS rates, birth rates, and urban-rural settings. Outcome measures should extend beyond mode of birth to include maternal and neonatal safety outcomes, women's and health workers' perspectives, and cost-effectiveness. As shown in Fig 4, financial interventions rarely reported safety outcomes compared to regulatory interventions. Ensuring that reductions in CS rate do not increase harm to women and babies is essential; thus, future financial intervention studies should evaluate maternal and neonatal health outcomes, care quality, and safeguards against unintended risks such as delayed medically necessary CS.

Evidence on the economic implications of regulatory strategies also remains limited. Only one complex regulatory intervention study reported cost data, and the intervention was not cost-effective. To inform policy and enable comparison across studies, future evaluations should include standardised and transparent reporting of costs and cost-effectiveness outcomes. More interventional research is needed in LMIC settings, as drivers of CS overuse and underuse may differ from high-income countries, which may limit the transferability of evidence from interventional research conducted in high-income countries. For example, limited surgical capacity, inequitable access, and variable clinical governance may influence the feasibility and impact of interventions.

Finally, few studies meaningfully involved women and health workers in intervention design and evaluation, despite these being key features of successful complex regulatory interventions. Given the strong influence of socioeconomic and cultural factors on CS rates, multidisciplinary collaboration among policymakers, health workers, women, and community is essential [66,67]. Engaging these groups throughout design and implementation can improve adoption, adaptability, and sustainability of interventions [15,69,70]. Furthermore, the use of theoretical or behaviour change frameworks was rare, despite their potential to explain the complex drivers of CS and guide the design of feasible, acceptable, and effective interventions [71,72]. Future research should integrate such frameworks to strengthen intervention design, implementation, and impact.

### Strengths and limitations of the study

This review provides new and up-to-date advancements on financial and/or regulatory interventions to reduce CS. We conducted a sibling study search to obtain comprehensive information about the interventions and applied WHO frameworks to ensure rigour in defining and categorising the interventions. A bubble chart was used to visualise and compare

the relative impact of different intervention types. Despite these strengths, this review has some limitations. The studies varied in design and outcome types and generally had low quality. Thus, it was not possible to conduct a pooled analysis, and all were assessed as low or very low certainty, so effectiveness should be interpreted with caution. Given the small number of studies by intervention type, we cannot ascertain which specific interventions are most effective for each intervention type. Furthermore, as health system interventions become more complex and multi-component, it becomes increasingly challenging to isolate effects and rigorously measure impact, raising concerns around measurability bias. This review did not compare intervention effects across low-, middle-, and high-income settings, as the included studies were highly heterogeneous in design, context, and intervention type. Lastly, this review did not include studies aimed at increasing CS use, which typically address barriers in under-resourced settings. While important, these strategies operate within a different policy and clinical context and were beyond the scope of this review.

## Conclusion

Despite the paucity of high-quality studies, current evidence allows for several important observations. Complex regulatory interventions that combine policy mandates with accountability mechanisms, health workers' training and guidance, incentives or penalties, women's engagement, and system-level coordination tend to report greater possible benefits in reducing CS while maintaining safety, though they may be resource-intensive. Financial reforms that shift from fee-for-service to alternative payment models (e.g., DRGs) also showed more possible benefits in reducing CS rates than simply equalising fees between vaginal births and CS, yet evidence on safety remains limited. These regulatory and financial interventions must be supported by regulatory measures such as clinical audits to ensure the quality and safety. Unnecessary CS is influenced by both demand-side factors (e.g., women's preferences) and supply-side factors (e.g., provider behaviour, financial incentives, and regulatory frameworks). Because these drivers and system dynamics evolve over time, both financial and regulatory interventions must be embedded within an adaptive, agile health system that continuously monitors, evaluates, and adjusts policies to sustain safe, equitable, and effective care. Future research should focus on evaluating context-specific complex interventions that consider safety outcomes, women's choice, and employ rigorous and ethical study designs, and articulate the underlying theory of change to explain why effectiveness varies across settings.

## Supporting information

**S1 Appendix. Preferred Reporting Items for Systematic reviews and Meta-Analyses extension for Scoping Reviews (PRISMA-ScR) Checklist.** From: Tricco AC, Lillie E, Zarin W, O'Brien KK, Colquhoun H, Levac D, et al. PRISMA Extension for Scoping Reviews (PRISMAScR): Checklist and Explanation. Ann Intern Med. 2018;169:467–473. https://doi.org/10.7326/M18-0850. Page M J, McKenzie J E, Bossuyt P M, Boutron I, Hoffmann T C, Mulrow C D et al. The PRISMA 2020 statement: an updated guideline for reporting systematic reviews BMJ 2021; 372:n71 https://doi.org/10.1136/bmj.n7. Page M J, Moher D, Bossuyt P M, Boutron I, Hoffmann T C, Mulrow C D et al. PRISMA 2020 explanation and elaboration: updated guidance and exemplars for reporting systematic reviews BMJ 2021; 372:n160 https://doi.org/10.1136/bmj.n160. (DOCX)

**S2 Appendix. Type of interventions, mapped to the WHO health system building blocks.** (DOCX)

**S3 Appendix. Search strategy.** (DOCX)

**S4 Appendix. Data extraction form.** (XLSX)

**S5 Appendix. Critical appraisal and certainty assessment results.**
(DOCX)

**S6 Appendix. Outcomes and impact reported for each study.**
(DOCX)

**S7 Appendix. Characteristics of included studies.**
(DOCX)

## Author contributions

**Conceptualization:** Rana Islamiah Zahroh, Alya Hazfiarini, Martha Vazquez Corona, Thiago Melo Santos, Nicole Minckas, Newton Opiyo, Fahdi Dkhimi, Veloshnee Govender, Meghan A. Bohren, Ana Pilar Betrán.

**Data curation:** Rana Islamiah Zahroh, Alya Hazfiarini, Martha Vazquez Corona, Nicole Minckas, Newton Opiyo.

**Formal analysis:** Rana Islamiah Zahroh, Alya Hazfiarini, Martha Vazquez Corona, Thiago Melo Santos, Nicole Minckas, Fahdi Dkhimi, Veloshnee Govender, Meghan A. Bohren, Ana Pilar Betrán.

**Funding acquisition:** Fahdi Dkhimi, Veloshnee Govender, Meghan A. Bohren, Ana Pilar Betrán.

**Investigation:** Rana Islamiah Zahroh, Thiago Melo Santos, Nicole Minckas, Newton Opiyo, Fahdi Dkhimi, Veloshnee Govender, Meghan A. Bohren, Ana Pilar Betrán.

**Methodology:** Rana Islamiah Zahroh, Thiago Melo Santos, Nicole Minckas, Newton Opiyo, Fahdi Dkhimi, Veloshnee Govender, Meghan A. Bohren, Ana Pilar Betrán.

**Project administration:** Rana Islamiah Zahroh.

**Software:** Rana Islamiah Zahroh.

**Supervision:** Nicole Minckas, Meghan A. Bohren, Ana Pilar Betrán.

**Validation:** Alya Hazfiarini, Martha Vazquez Corona.

**Visualization:** Thiago Melo Santos.

**Writing – original draft:** Rana Islamiah Zahroh.

**Writing – review & editing:** Alya Hazfiarini, Martha Vazquez Corona, Thiago Melo Santos, Nicole Minckas, Newton Opiyo, Fahdi Dkhimi, Veloshnee Govender, Meghan A. Bohren, Ana Pilar Betrán.

## Acknowledgments

We extend our thanks to Reihaneh Moniri for double reviewing the extraction of Persian studies, and Patrick Condron and Dr Steve McDonald for their support on the search strategy update.

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
