## [Decision Letter · Decision Letter 0]

17 Sep 2025

PGPH-D-25-02141

Financial and regulatory interventions to reduce unnecessary caesarean sections: an updated scoping review

Dear Dr. Zahroh,

Thank you for submitting your manuscript to PLOS Global Public Health. After careful consideration, we feel that it has merit but does not fully meet PLOS Global Public Health’s publication criteria as it currently stands. Therefore, we invite you to submit a revised version of the manuscript that addresses the points raised during the review process.

We look forward to receiving your revised manuscript.

Kind regards,

Jerin Jose Cherian, M.D.

Academic Editor

Journal Requirements:

1. Please provide a detailed online Financial Disclosure statement. This is published with the article. It must therefore be completed in full sentences and contain the exact wording you wish to be published.

a) Please clarify all sources of financial support for your study. List the grants, grant numbers, and organizations that funded your study, including funding received from your institution. Please note that suppliers of material support, including research materials, should be recognized in the Acknowledgements section rather than in the Financial Disclosure.

b) State the initials, alongside each funding source, of each author to receive each grant. For example: “This work was supported by the National Institutes of Health (####### to AM; ###### to CJ) and the National Science Foundation (###### to AM).”

c) State what role the funders took in the study. If the funders had no role in your study, please state: “The funders had no role in study design, data collection and analysis, decision to publish, or preparation of the manuscript.”

For more information, please go to our submission guidelines:

https://journals.plos.org/globalpublichealth/s/submission-guidelines#loc-financial-disclosure-statement

2. Please ensure that the funders and grant numbers match between the Financial Disclosure field and the Funding Information tab in your submission form. Note that the funders must be provided in the same order in both places as well.

3. Please update your online Competing Interests statement. If you have no competing interests to declare, please state: “The authors have declared that no competing interests exist.”

Additional Editor Comments:

The manuscript is extremely relevant and deals with an interesting theme. There are however some comments that we share with you from multiple reviewers. As you can see, we could gather interest from 6 reviewers for your manuscript, which is indicative of the relevance of your manuscript. We admit that there might be some overlap and perhaps even some minor conflicts in the recommendation, but request you to address them in the tabular response to comments document. We are confident that addressing these comments will improve the quality of your manuscript significantly.

Reviewers' comments:

Reviewer's Responses to Questions

**Comments to the Author**

1. Does this manuscript meet PLOS Global Public Health’s publication criteria?

Reviewer #1: Yes

Reviewer #2: Yes

Reviewer #3: Partly

Reviewer #4: Yes

Reviewer #5: Yes

Reviewer #6: Yes

2. Has the statistical analysis been performed appropriately and rigorously?

Reviewer #1: N/A

Reviewer #2: N/A

Reviewer #3: Yes

Reviewer #4: N/A

Reviewer #5: No

Reviewer #6: N/A

3. Have the authors made all data underlying the findings in their manuscript fully available (please refer to the Data Availability Statement at the start of the manuscript PDF file)?

Reviewer #1: No

Reviewer #2: Yes

Reviewer #3: Yes

Reviewer #4: Yes

Reviewer #5: Yes

Reviewer #6: Yes

4. Is the manuscript presented in an intelligible fashion and written in standard English?

Reviewer #1: Yes

Reviewer #2: Yes

Reviewer #3: Yes

Reviewer #4: Yes

Reviewer #5: Yes

Reviewer #6: Yes

Reviewer #1: Good work! A lot of effort went into this.

1. Please add date accessed for reference 2, 12 & add specific link to the file.

2. This is a scoping review. I'm not sure why you would exclude studies aiming to increase CS use. While, we agree that unnecessary CS use should be avoided, it is a missed opportunity to critically evaluate the motivations behind centres who want to increase CS use. Knowledge translation is ineffective without understanding the opposing side of the story. (Line 122). If you want to argue significant knowledge addition since Opiyo et al (2020) was published, it would have been ideal to include those papers.

3. It's been a year since your last search strategy. You had almost 13k papers as such I will not advise an updated search at this time. However, this comment can't supersede a request for updated search by another reviewer. Next time be more specific with your search to remove case reports and other exclusion studies upfront.

4. Next time when reviewing "sibling studies", please include both forward and backward citations. Not just forward citations. Use a tool like "Citation Chaser" to help you get the lists. Upload the list to Covidence afterwards.

5. You mention selection criteria for "sibling studies" (i.e., double check by another reviewer) but I do not see the exact screening criteria for main studies in the methodology. Please re-state it and don't assume the reader will be a scoping review expert to know this step.

6. Include the data extraction form in the appendix (if not included already) and mention it in the methods.

7. Next time use ROBINS-I for risk of bias analysis. RoB 2 is mostly for Clinical Trials but is acceptable here as a risk of bias is not necessary per se in scoping reviews. You didn't list the # of papers that evaluate policies. I'm not sure you can even do RoB 2 on those.

8. I do not have the energy to open Opiyo et al (2020) to review the risk of bias for the remaining studies. Combine and include them here in your table appendix. Are you even serious with this comment: "The remaining study’s quality appraisal can be seen on Opiyo et al (2020) review?." A very privileged statement expecting readers to find the access, time to download, review the other paper, whose first author is a co-author on this paper.

9. It's very hard to read the results. You have regurgitated your data extraction sheet word for word. Can you make a few comparative statements between papers and improve the story a bit more with adding interpretation. Cut down the text in the result section by 10-25%. Please, this is supposed to be a journal article, not your dissertation :).

10.This suggests that some interventions may lack the specificity or sensitivity needed to distinguish between necessary and unnecessary CS. Can you list all interventions that have a high vs low specificity and sensitivity so we get a take home message of what might work and what might not?

11. Cut down discussion by 10-25% and limit conclusion to one paragraph.

Reviewer #2: The methodology of this work is well-documented and utilize a systematic approach and an exhaustive search process. The detailed systematic review of such a relevant topic (the impact of regulatory and financial interventions on the reduction of unnecessary cesarean) is highly valued.

This exploratory study contributes to the scientific literature by providing an updated review of the 2020 work by Opiyo et al. It accurately demonstrates that while new studies have emerged, there has been no significant advancement in the quality of the evidence base, thus contributing to the reaffirmation of the need for additional scientific research, utilizing consistent scientific methodologies to generate high-quality evidence.

It is recommended that the discussion of the results be cautious, given that it is based on studies with a low level of scientific evidence.

The lack of robust scientific studies, particularly in low- and middle-income countries (where the largest increases in CS rates have been observed), limits the possibility of conducting a meta-analysis or presenting novel scientific findings. In this sense, the novel scientific contribution of the work may require more years with additional studies.

To strengthen the scientific contribution of the work, it is recommended that the author provides a rigorous justification and a detailed rationale for how the obtained results contribute to the current knowledge in the field.

Reviewer #3: Your article is a timely and important update on financial and regulatory interventions to address the world's global cesarean delivery epidemic. It beautifully builds on and extends the 2020 scoping review by Opiyo et al., doubling the number of evaluations and providing a structured classification of interventions using WHO frameworks. Key strengths I identified include protocol registration, adherence to PRISMA-ScR, inclusion of sibling studies, and the balanced discussion of potential harms. Nonetheless, I recommend some key revisions to strengthen methodological clarity and also address some overstated conclusions. These include:

1) Please clarify and correct the mis-labelling of RoB tools (RoB 2 vs ROBINS-I), to address the risk of methodological bias (Page 7, lines 168–176.)

2) Please clarify the inclusion of pre-2019 studies compared to the stated 2019–2024 window, and then update PRISMA accordingly (Page 6, lines 137–144; also Page 11, lines 211–215).

3) In the Abstract and Discussion statements the paper seems to overstate the strength of its claims which do not align with the stated low/very-low certainty evidence (Page 2, lines 38–42; Page 17, lines 365–373.).

4) It is important to provide the article's rationale for applying GRADE in a scoping review (typically expected in a Systematic Reviews), or otherwise move this to the supplementary files section.

5) On key definitions for the study population (persons undergoing cesarean deliveries), please clarify the use of “low-risk” when including Robson Group 5 as this doe not seem to tie with the WHO Robson framework definitions (Page 5, lines 98–101.).

6) I propose to adopt an equity and generalisability lens to expand discussion on the implications for LMICs, where finances and regulatory interventions could underlie the coexistence of CS underuse and overuse.

7) The article could highlight the lack of cost-effectiveness evidence and recommend systematic cost reporting in future evaluations.

Other minor improvements include addressing some redundancy in the Discussion section.

Overall, with these revisions, this paper has the potential to make a valuable contribution to policy and practice in global maternal health.

Reviewer #4: 1. In Table 1, consider adding examples from included studies to illustrate categories (e.g., link to specific interventions) for better understanding. Also, the categories in this table would be more useful if the results are aligned with them.

2. Audits are described under Contracting modalities including reporting obligations as well as Ensuring accountability.

3. Table 2 only presents terminology for low-certainty evidence, even though the inclusion criteria encompassed RCTs and quasi-experimental designs. As readers may expect at least some interventions to be evaluated in RCTs (which under GRADE would begin at high certainty), it would be important for the authors to explain why the table is limited to low certainty as this is definition and not based on results.

4. Table 2 outlines standardised terminology for interpreting evidence certainty (e.g., ‘may reduce,’ ‘may have no effect’), yet this phrasing is not consistently reflected in the Results or Discussion, where findings are reported narratively.

5. The results are largely narrative. Consider including descriptive, study-level effect metrics such as absolute and relative changes in CS rates, reported RRs/ORs with 95% Cis etc, presented in tables or figures by intervention category.

Reviewer #5: REVIEWER FEEDBACK

SUMMARY OF FEEDBACK

This is a very interesting paper. Globally, there has been rise in caesarean section (CS)rates. Most of these rates are unnecessary and a cost on health systems. Interventions that can be implemented to reduce on the rates will tend to be welcome in most settings. Financial and regulatory interventions to reduce unnecessary caesarean sections have been implemented in a number of settings. However, most of the evidence on their effectiveness is scanty, inconclusive or not reliable.

The study has rightly pointed out reducing caesarean sections requires a combination of both financial and regulatory strategies. This is to ensure that there is a balance on approaches and avoid provider preferences if financial is considered only in absence of regulation.

Abstract

Line 28-29. The first sentence under methods should ideally be last sentence under introduction. It gives reader an idea of the aim of the study.

Results:

Line 33. Did the 15 sibling studies included enhance scoping review of the subject? How did it do that?

Introduction

I suggest that you consider defining what caesarean section rate is so that a reader can have clear understanding of the issue.

Line 49-50. Consider providing the acceptable thresholds for CS, as guided by World Health Organization to appreciate statement that “global rates of CS are rising beyond levels justified by clinical need”

Line 54. Provide common examples of adverse maternal and neonatal outcomes linked with “unnecessary” CS

Line 55. Do the supply-side health system factors happen to be drivers of high CS rates for both elective and non-elective cases or there is a distinction?

Line 65-67. The sentence is quite interesting. WHO has noted that largest increase in CS rates is in low-and-middle-income countries? However, the study reports that most of the papers reviewed were from high-income countries. It would be nice if any of the reviewed papers was from the low-income countries to appreciate this statement.

Line 70-75. These citations are great. However, provide a summary of what they found.

Line 84-85. Consider providing clear explanation on the indicators to ascertain effectiveness, safety and outcomes for the study.

Line 132-134. Mention examples of indicators on effectiveness, safety and outcomes that you mapped. Be very clear.

Line 139-141. What was the rationale for including papers excluded by Opiyo et al (2020)? By re-screening these papers, did you find something omitted that added to this current paper?

Line 152-159. Consider providing clear process used in including or excluding studies for review. How did the two authors choose which paper was to be included and was not to be included?

Line 166-167. How was testing for data extraction done? How was it reviewed by all the authors? How did we ensure that data extraction was sound and appropriate for the task?

Line 218-221. How many included studies were from low-and-middle-income countries, if any? How was the picture on the indicators of interest on effectiveness, safety and outcomes from the low-and-middle-income countries compared to the high-income countries studies?

Line 233. Consider mentioning the theoretical and behaviour change frameworks used during intervention design for these two studies reviewed.

Conclusion

The conclusion should reflect on the aim of the review and whether it has been achieved. Consider strengthening your conclusion based on great findings reported.

Reviewer #6: 1. The manuscript is described as a scoping review, but parts of the analysis (e.g., GRADE certainty ratings) resemble a systematic review. The authors should clarify the intended scope and avoid overstating certainty.

2. Most interventions are from high-income countries, yet the fastest rise in CS rates is in LMICs. The discussion should more explicitly address how findings translate to LMIC settings and what gaps remain.

3. The results are detailed but sometimes difficult to follow. A concise summary table or figure comparing intervention type, setting, outcomes, and certainty of evidence would make the findings more accessible for policymakers and practitioners.

4. The manuscript notes adverse outcomes (e.g., postpartum haemorrhage, infection, neonatal respiratory issues) but does not explore mechanisms or propose mitigation strategies. A deeper analysis of how to prevent or monitor these harms is needed.

5. While the policy implications are well stated, recommendations could be more specific on:

Which types of interventions are most feasible in different health system contexts.

How policymakers can balance financial reforms with regulatory safeguards.

Strategies for engaging women and health workers in intervention design.

**Do you want your identity to be public for this peer review?** For information about this choice, including consent withdrawal, please see our Privacy Policy

Reviewer #1: No

Reviewer #2: **Yes:** Maria Elena Critto

Reviewer #3: **Yes:** SAMUEL AKOMBENG OJONG

Reviewer #4: No

Reviewer #5: **Yes:** Benedictus Mangala

Reviewer #6: **Yes:** DR D M SHILPA

---

## [Decision Letter · Decision Letter 1]

8 Dec 2025

PGPH-D-25-02141R1

Financial and regulatory interventions to reduce unnecessary caesarean sections: an updated scoping review

Dear Dr. Zahroh,

Thank you for submitting your manuscript to PLOS Global Public Health. After careful consideration, we feel that it has merit but does not fully meet PLOS Global Public Health’s publication criteria as it currently stands. Therefore, we invite you to submit a revised version of the manuscript that addresses the points raised during the review process.

Please address the four comments provided by the reviewers in your results and discussion sections, that will help improve the quality and readability of the publication.

We look forward to receiving your revised manuscript.

Kind regards,

Jerin Jose Cherian, M.D.

Academic Editor

Journal Requirements:

Additional Editor Comments (if provided):

Reviewers' comments:

Reviewer's Responses to Questions

**Comments to the Author**

Reviewer #4: (No Response)

Reviewer #6: All comments have been addressed

publication criteria?

Reviewer #4: Yes

Reviewer #6: Yes

3. Has the statistical analysis been performed appropriately and rigorously?

Reviewer #4: N/A

Reviewer #6: N/A

4. Have the authors made all data underlying the findings in their manuscript fully available (please refer to the Data Availability Statement at the start of the manuscript PDF file)?

Reviewer #4: (No Response)

Reviewer #6: Yes

5. Is the manuscript presented in an intelligible fashion and written in standard English?

Reviewer #4: Yes

Reviewer #6: Yes

Reviewer #4: 1. The organization of Results by intervention type followed by simple/complex classification is appropriate, but operational details remain too minimal for readers to understand the practical content of interventions. Adding 1-2 sentences describing what was actually changed (payment rule, legal mechanism etc.) would improve interpretability and better reflect the aim of the scoping review.

2. The narrative reads like sequential study summaries rather than a mapped evidence landscape. Can add synthesized intervention outcome pattern statements.

3. Linking intervention descriptions more clearly to their intended mechanism for reducing unnecessary CS can add to the current structure

4. Consider adding a summary table or infographic that clearly maps the intervention types and sub-types (24) alongside their reported direction of effect on caesarean-section rates, as this would greatly enhance clarity and allow readers to quickly grasp the evidence landscape.

Reviewer #6: The authors have address all the questions posed by the reviewers and hence can be considered for further process.

**Do you want your identity to be public for this peer review?** For information about this choice, including consent withdrawal, please see our Privacy Policy

Reviewer #4: No

Reviewer #6: **Yes:** Dr D M Shilpa

---

## [Editor Report · Decision Letter 2]

29 Dec 2025

Financial and regulatory interventions to reduce unnecessary caesarean sections: an updated scoping review

PGPH-D-25-02141R2

Dear Dr Zahroh,

We are pleased to inform you that your manuscript 'Financial and regulatory interventions to reduce unnecessary caesarean sections: an updated scoping review' has been provisionally accepted for publication in PLOS Global Public Health.

Best regards,

Jerin Jose Cherian, M.D.

Academic Editor